# Evaluation of the Antinociceptive, Antiallodynic, Antihyperalgesic and Anti-Inflammatory Effect of Polyalthic Acid

**DOI:** 10.3390/molecules26102921

**Published:** 2021-05-14

**Authors:** Juan Rodríguez-Silverio, María Elena Sánchez-Mendoza, Héctor Isaac Rocha-González, Juan Gerardo Reyes-García, Francisco Javier Flores-Murrieta, Yaraset López-Lorenzo, Geovanna Nallely Quiñonez-Bastidas, Jesús Arrieta

**Affiliations:** 1Escuela Superior de Medicina, Instituto Politécnico Nacional, Plan de San Luis y Díaz Mirón, Colonia Casco de Santo Tomás, Miguel Hidalgo, Ciudad de México 11340, Mexico; jrsilverio61@yahoo.com.mx (J.R.-S.); mesmendoza@hotmail.com (M.E.S.-M.); heisaac2013@hotmail.com (H.I.R.-G.); juangreyesgarcia@gmail.com (J.G.R.-G.); fjfloresmurrieta@yahoo.com.mx (F.J.F.-M.); yarlop_2310@outlook.com (Y.L.-L.); geovanna_quinonez@hotmail.com (G.N.Q.-B.); 2Unidad de Investigación en Farmacología, Instituto Nacional de Enfermedades Respiratorias Ismael Cosió Villegas, Secretaría de Salud, Ciudad de México 14080, Mexico

**Keywords:** polyalthic acid, naproxen, isobologram, antinociception, antiallodynic effect, synergism

## Abstract

Nonsteroidal anti-inflammatory drugs (NSAIDs) are very commonly used, but their adverse effects warrant investigating new therapeutic alternatives. Polyalthic acid, a labdane-type diterpenoid, is known to produce gastroprotection, tracheal smooth muscle relaxation, and antitumoral, antiparasitic and antibacterial activity. This study aimed to evaluate the antinociceptive, antiallodynic, antihyperalgesic and anti-inflammatory effect of polyalthic acid on rats. Moreover, the effectiveness of treating hyperalgesia with a combination of polyalthic acid and naproxen was analyzed, as well as the type of drug–drug interaction involved. Nociception was examined by injecting 1% formalin into the right hind paw and thermal hyperalgesia and inflammation by injecting a 1% carrageenan solution into the left hind paw of rats. Allodynia was assessed on an L5/L6 spinal nerve ligation model. Polyalthic acid generated significant antinociceptive (56–320 mg/kg), antiallodynic (100–562 mg/kg), and antihyperalgesic and anti-inflammatory (10–178 mg/kg) effects. Antinociception mechanisms were explored by pretreating the rats with naltrexone, ODQ and methiothepin, finding the effect blocked by the former two compounds, which indicates the participation of opioid receptors and guanylate cyclase. An isobolographic analysis suggests synergism between polyalthic acid and naproxen in the combined treatment of hyperalgesia.

## 1. Introduction

Pain is the most common symptom by which diseases manifest themselves. The main function of pain is to protect the integrity of the organism against potentially damaging factors. The same painful stimulus can be perceived in different intensities (mild, moderate, intense) by distinct individuals, depending on psychological, social and cultural factors, among others. Pain perception can become pathological if there is an imbalance in the mechanisms involved [1,2].

For treating pain, nonsteroidal anti-inflammatory drugs (NSAIDs) are among the most widely used medications [3]. When frequently taken, however, NSAIDs can trigger adverse events, such as gastrointestinal ulcers (with consequent bleeding, perforation and/or obstruction), kidney dysfunction, and cardiovascular events, all entailing the risk of death [4]. Although NSAIDs that are selective cyclooxygenase-2 inhibitors (Coxibs) reduce the incidence of gastrointestinal complications, they have been linked to kidney problems and an increased risk of cardiovascular complications [5,6,7]. Thus, new types of pain management therapies are needed.

Medicinal plants are a possible alternative for treating pain. Indeed, some of the metabolites derived from plants and employed in traditional medicine have provided the basis for discovering and developing modern drug therapy [8]. Labdane-type diterpenes (Figure 1A) are an excellent example of natural products with analgesic activity. They also have a potential antifungal, antibacterial, antimutagenic, cytotoxic and anti-inflammatory effect [9]. Polyalthic acid is a labdane-type diterpenoid (Figure 1B) known to promote gastroprotection [10] and the relaxation of tracheal smooth muscle cells [11], as well as having antitumoral [12], antiparasitic [13] and antibacterial activity [14]. The anti-inflammatory and analgesic potential of polyalthic acid has not yet been described in the literature to the best of our knowledge. Hence, the current contribution aimed to evaluate the antinociceptive, antiallodynic, antihyperalgesic and anti-inflammatory effect of polyalthic acid on rats with models of pain and inflammation. For the combined treatment of hyperalgesia with polyalthic acid and naproxen, moreover, the type of drug–drug interaction was analyzed.

## 2. Results

### 2.1. Antinociceptive Effect of Polyalthic Acid

Oral administration of polyalthic acid to rats led to a significant decrease in nociception. The latter was measured as the number of flinches of the right hind paw of rats, the site where 1% formalin was injected (Figure 2A). Moreover, a dose-dependent antinociceptive activity was found on the formalin test when the compound was administered at 56 to 320 mg/kg (Figure 2B). The 178 and 320 mg/kg doses afforded a 45.2 and 56.3% antinociceptive effect, respectively, which was slightly higher than the 32.4% antinociceptive effect provided by naproxen (the reference drug, evaluated at 32 mg/kg; Figure 2B). These three treatments were significantly different from the vehicle-treated control.

### 2.2. Possible Mechanisms of the Antinociceptive Effect of Polyalthic Acid

The antinociceptive activity generated by polyalthic acid (320 mg/kg) orally administered to rats was significantly impeded by the intraperitoneal administration of a non-selective opioid receptor antagonist, naltrexone (Figure 3A, 1 mg/kg), and by a selective inhibitor of nitric oxide-sensitive guanylyl cyclase, ODQ (Figure 3C, 2 mg/kg). Contrarily, methiothepin (0.1 mg/kg), a selective antagonist of 5HT1,5 receptors, did not affect the antinociception of polyalthic acid (Figure 3B). The administration of naltrexone, methiothepin or ODQ did not cause any effects on the nociceptive behavior of rats, per se.

### 2.3. Antiallodynic Effect of Polyalthic Acid on Rats with an L5/L6 Spinal Nerve Ligation

Rats that underwent an L5/L6 spinal nerve ligation became more sensitive to nociception, evidenced by the reduced 50% withdrawal threshold to a value less than 4 g. This greater sensitivity was interpreted as tactile allodynia. Sham surgery had no significant impact on the results. Orally administered polyalthic acid increased the 50% withdrawal threshold in allodynic rats (Figure 4A). The administration of polyalthic acid at 100 to 562 mg/kg generated a significant and dose-dependent antiallodynic effect (Figure 4B), decreasing allodynia by a minimum of 24.63% (at 100 mg/kg) and a maximum of 70.7% (at 562 mg/kg). The activity at the highest dose (562 mg/kg) was not significantly different from the 75.3% effect produced by pregabalin (the reference drug, at 10 mg/kg; Figure 4B), which was also significantly different compared to the data of the control group. Whereas the activity of polyalthic acid was null at 24 h post-administration, pregabalin was only reduced at this time.

### 2.4. Antihyperalgesic Effect of Polyalthic Acid and Naproxen

The subcutaneous injection of 1% carrageenan into the left hind paw of rats caused inflammation and a decrease in latency to the withdrawal of the paw from a heated surface (30 °C), considered as thermal hyperalgesia. The effect persisted for at least 6 h. Polyalthic acid (178 mg/kg) and naproxen (32 mg/kg) administered individually showed antihyperalgesic activity (Figure 5A,B), which was dose-dependent for both compounds, administered at 10–178 mg/kg and 1–32 mg/kg, respectively (Figure 5C,D). The maximum antihyperalgesic effect was 65.50% for polyalthic acid (at 178 mg/kg) and 58.49% for naproxen (at 32 mg/kg). On the other hand, the effective dose resulting in a 50% reduction (ED_50_) in the hyperalgesic response (compared to the control group) was 101.6 ± 30.9 mg/kg for polyalthic acid and 20.1 ± 4.8 mg/kg for naproxen.

### 2.5. Synergistic Interaction between Polyalthic Acid and Naproxen

Once the theoretical ED_50_ was calculated (60.9 ± 15.7), a dose–response curve was constructed for effect produced by the five different doses of the combination treatment of polyalthic acid plus naproxen (for details, see Table 1). Based on the 1% carrageenan test, hyperalgesia was significantly diminished in a dose-dependent fashion (Figure 6A). The combination treatment with the highest dose (50.8 mg/kg polyalthic acid + 10 mg/kg naproxen) showed almost complete elimination of carrageenan-induced hyperalgesia, being 96.07% compared to the vehicle-treated control (Figure 6A).

The experimental ED_50_ (2.4 ± 0.02 mg/kg) was calculated from the dose–response curve of the combination treatment with polyalthic acid and naproxen (Table 2). As can be appreciated in the isobologram, the point on the graph for the experimental ED_50_ is located well below the theoretical additive line of the combination treatment (Figure 6B). Therefore, a synergistic interaction is suggested. Additionally, the Student’s *t*-test established a significant difference between the theoretical (60.9 ± 15.7 mg/kg) and experimental (2.4 ± 0.02 mg/kg) ED_50_ of the combination treatment (Table 2). Finally, the interaction index (γ = 0.04 ± 0.02) was <1, confirming a synergistic interaction between polyalthic acid and naproxen in the combination treatment.

### 2.6. Anti-Inflammatory Effect of Polyalthic Acid

The subcutaneous injection of 1% carrageenan into the left hind paw of rats generated edema that persisted during the 6 h experiment (Figure 7A). All doses of polyalthic acid (10 to 178 mg/kg) significantly inhibited the edema in a dose-dependent fashion (Figure 7B). The most limited anti-inflammatory effect (29.86%) was provided by the lowest dose (10 mg/kg). The 70.9 and 82.4% effect produced by the 100 and 178 mg/kg doses of polyalthic acid, respectively, were significantly higher than the 47.4% result achieved with the 32 mg/kg dose of naproxen (Figure 7B), which in turn was significantly different from the outcome of the vehicle treatment (the control).

## 3. Discussion

An evaluation was made of the antinociceptive, antiallodynic, antihyperalgesic and anti-inflammatory effect of polyalthic acid orally administered to female rats. These animals showed a greater inflammation response [15] and a higher pain threshold [16] than male rats in carrageenan-induced inflammation and hyperalgesia, respectively. Meanwhile, there are no reported differences in the results on the formalin test [17,18,19,20] or with L5-L6 spinal nerve ligation [21] between female and male rats. The possibility that the effectiveness of polyalthic acid is distinct from male rats should be explored in future studies.

To our knowledge, though there are descriptions in the literature of polyalthic acid’s antiulcer [10], antitumoral [12], and antipathogenic effects [13], this is the first report of its antinociceptive activity. Polyalthic acid presently provided dose-dependent antinociception, including a significantly reduced amount of time spent on licking behavior in phase II of the formalin test. This phase, involving both inflammatory mechanisms and central nervous system sensitization, is known to respond to various drugs with established clinical analgesic action. Polyalthic acid was more effective than naproxen, evaluated at a maximum dose of 32 mg/kg.

Based on the current data from the formalin test on rats, the intraperitoneal administration of naltrexone and ODQ, but not methiothepin, decreased the antinociceptive activity of polyalthic acid. Since naltrexone is an opioid antagonist, its capacity to affect nociception reveals the activation of opioid receptors by polyalthic acid. The ODQ-induced change indicates that cyclic guanosine monophosphate (cGMP) was activated by polyalthic acid. No information on the mechanism of action of polyalthic acid was found in the literature concerning an in vivo rat model.

According to a previous study, the relaxing effect of polyalthic acid on guinea-pig isolated trachea rings was not mediated by the activation of nitric oxide or K^+^ channels [11]. Thus, further experiments are necessary to clarify the mechanism of action of this compound. Notwithstanding, the current findings show the activation of cyclic guanylyl cyclase by polyalthic acid. Nitric oxide is known to activate guanylyl cyclase, which in turn stimulates the production of cGMP and subsequently activates K^+^ channels [22]. Considering that the administration of a membrane-permeable analog of cGMP generates antinociception [23], membrane hyperpolarization is responsible for the effect. We hypothesize that the antinociceptive activity of polyalthic acid may stem from the activation of NO-cGMP-K^+^ channels, which are reported to have an important role in antinociception [24].

Regarding evidence of the participation of opioid receptors in the antinociceptive activity of polyalthic acid, an antinociceptive effect of opioids has been observed in several clinical [25] and experimental studies [26]. The opioid system is closely related to the nitric oxide pathway. The intra-hippocampal CA1 injection of L-NAME or L-arginine prevents the antinociceptive activity of the systemic administration of morphine [27]. Very relevant to the present results, peripheral administration of methylene blue, an inhibitor of guanylyl cyclase, prevents the antihyperalgesic effect induced by the local administration of morphine [28].

Based on this study and a previous report [29], activation of the opioid receptor pathway stimulates cGMP. Supporting this idea, an agonist of opioid receptors (morphine) is reported to activate the PI3γ/AKT/nNOS/NO/K^+^_ATP_ channel signaling pathway to provide antinociception [30,31]. The antinociception herein displayed by polyalthic acid could stem from three factors: (1) direct activation of cGMP; (2) indirect activation of cGMP through opioid receptors; or (3) activation of the NO-cyclic GMP-K^+^ channel pathway through opioid receptors.

The oral administration of polyalthic acid on the 14th day after the induction of neuropathic pain caused a reduction of tactile allodynia in a dose-dependent manner. Polyalthic acid was less potent than the reference drug pregabalin. To obtain the same antinociception effect, the dose required was higher for polyalthic acid than pregabalin. Hence, further preclinical research is necessary to determine whether the chronic administration of polyalthic acid can help to reduce the dose required and/or whether its activity can be increased when administered with other drugs.

In the clinic, neuropathic pain is difficult to manage, and its control is usually refractory to opioid treatment, requiring high doses or more than one drug [25]. Hence, the structure of polyalthic acid might be useful for developing new drugs to treat neuropathic pain.

On the other hand, polyalthic acid and naproxen (a derivative of propionic acid) showed an antihyperalgesic effect in rats. The anti-inflammatory and analgesic properties of naproxen have been described, with the main adverse effects of gastric damage [32].

Like other NSAIDs, naproxen can inhibit the cyclooxygenase enzymes (COX1 and COX2), which have a pivotal role in producing prostaglandins and, therefore, the ensuing activation of cellular pathways involved in inflammation, nociception, and cytoprotection [33]. Thus, a reduced level of prostacyclin is involved in anti-inflammatory and antinociceptive effects as well as in gastric damage. Prostacyclin is the main cyclooxygenase product responsible for generating the protective gastric mucosal layer [34].

The five doses herein tested of the combination treatment of polyalthic acid plus naproxen all afforded significant antihyperalgesic activity. A dose–response curve was constructed to determine the ED_50_ of the polyalthic acid–naproxen combination, and an isobolographic analysis was carried out to identify the type of interaction between these two compounds. The latter graph was based on the theoretical ED_50_ of polyalthic acid and naproxen when administered individually. The location of the experimental value of the combination treatment on the isobologram revealed a synergistic interaction between polyalthic acid and naproxen, which was confirmed by the interaction index. Indeed, naproxen is known to interact synergistically with some other drugs to reduce inflammation, nociception and hyperalgesia [35].

We hypothesize that the synergistic interaction between naproxen and polyalthic acid might be due to pharmacodynamic drug–drug interactions caused by the supra-additive effects of distinct mechanisms of action. The current antihyperalgesic effect of polyalthic acid was mediated by the activation of opioid receptors and guanylyl cyclase. These two mechanisms of polyalthic acid are different from the antihyperalgesic mechanism of naproxen. Hence, the results strengthen the hypothesis of pharmacodynamics. Although no information was found in the literature about the pharmacokinetics of polyalthic acid, a pharmacokinetic interaction between this compound and naproxen is not discarded, and future research is needed in this regard.

Since many analgesic drugs exert anti-inflammatory activity simultaneously [36], the anti-inflammatory potential of polyalthic acid was presently evaluated with the carrageenan model, finding a significant reduction in paw edema. This effect may be related to the nitric oxide pathway apparently involved in the antinociceptive mechanism. There is increasing evidence of constant crosstalk between NO and COX at several levels, suggesting that NO can directly interfere with COX expression and PG biosynthesis. Therefore, medications that combine NO donors with NSAIDs have been synthesized and employed for treating inflammatory disorders [37].

## 4. Materials and Methods

### 4.1. Animals

All experiments were performed with female Wistar rats (180–200 g) acquired from our institution. The animals were placed in acrylic boxes with food and water provided ad libitum. They were maintained on a 12 h light/dark cycle in a controlled environment at room temperature (22 ± 1 °C) and with constant airflow. The animal protocols were conducted following the Guidelines on Ethical Standards for Investigation of Experimental Pain in Animals [38] and the Mexican norm (Technical Specification on the Production, Care and Use of Laboratory Animals, NOM-062-ZOO-1999) established by the Secretary of Agriculture (SAGARPA). They were approved by the Institutional Animal Care and Use Committee of the National Institute of Respiratory Diseases “Ismael Cosio Villegas” in Mexico City, Mexico (registration number B41-11). At the end of the experiments, the rats were sacrificed in a CO_2_ chamber.

### 4.2. Drugs

Polyalthic acid was obtained from the leaves of *Croton reflexifolius*, as previously described [11]. The leaves were collected in Tehuetlan, a town in the state of Hidalgo, Mexico. Purity was determined by comparison with an authentic sample in thin-layer chromatography and by determining the melting point. Polyalthic acid was administered in a suspension with 0.05% Tween-80. Naproxen sodium, pregabalin, methiothepin mesylate salt, and naltrexone were dissolved in 0.9% saline solution before use, while ODQ was dissolved in 25% DMSO. All drugs were purchased from Sigma-Aldrich (St. Louis, MO, USA).

### 4.3. Formalin Test

To assess the antinociceptive effect, rats were orally administered polyalthic acid (at different doses), naproxen (32 mg/kg) or the vehicle only (a saline solution with 0.05% of Tween-80) and submitted to the 1% formalin test [39] 1 h later. Thirty min before the test, each animal was placed in an open Plexiglas observation chamber to become acclimated to its surroundings. For the test, each animal was removed, subcutaneously injected with 50 μL of 1% formalin into the dorsal surface of the right hind paw, and immediately returned to its chamber. The quantification of nociceptive behavior was based on counting the number of flinches of the injected paw for 1 min and was performed every 5 min for 1 h. A flinch was defined as a rapid and brief withdrawal or flexing of the injected paw.

The behavior on the formalin test is biphasic. After the first phase of flinching behavior (0–10 min), there is a relatively short and quiescent period and then a prolonged tonic response (15–60 min). Data are expressed as the percentage of antinociceptive activity on the maximum number of flinches evoked in the vehicle-treated control during each phase. The percentage of antinociception was calculated with the following equation:$$\%\mathrm{Antinociception}=\frac{\mathrm{Control}\;\mathrm{licking}\;\mathrm{time}-\mathrm{Test}\;\mathrm{licking}\;\mathrm{time}}{\mathrm{Control}\;\mathrm{licking}\;\mathrm{time}}\times 100$$

### 4.4. L5/L6 Spinal Nerve Ligation

The effect of polyalthic acid on the neuropathic pain of rats was examined by ligating the L5 and L6 spinal nerve to induce allodynia [40]. Before surgery, animals were anesthetized with a mixture of ketamine/xylazine (45:12 mg/kg, i.p.). The incision site was prepared, and the dorsal vertebral column was exposed. The L5 and L6 spinal nerves were located and ligated with a 6–0 silk suture distal to the dorsal root ganglion. Sham-operated rats underwent the same anesthetization and surgical procedure but without nerve ligation. Upon completing the procedure, the incision was sutured, and the rats were individually housed for a 14 day recovery period. Animals with a motor deficiency were excluded.

Tactile allodynia was appraised by observing paw withdrawal in response to a series of calibrated von Frey filaments (Stoleting Co., Wood Dale, IL, USA), which exerted a force from 0.4 g (3.9 mN) to 15 g (147.1 mN). Withdrawal thresholds were determined by increasing and decreasing stimulus strength [41]. The stimulus intensity required to produce a response in 50% of the applications for each animal was defined as the 50% withdrawal threshold. The allodynia of all rats was verified before starting the experiment by assuring that they responded to a stimulus of less than 4 g. In all cases, a cutoff of 15 g was employed to avoid measurement mistakes. Testing was carried out immediately before administration and every hour up to 8 h afterward, with an additional measurement at 24 h. The vehicle, polyalthic acid (100, 178, 320 and 562 mg/kg) and pregabalin (positive control; 10 mg/kg) were all given orally. Data are presented as the percentage of MPE, which was calculated with the following equation:$$\%\mathrm{MPE}=\frac{\mathrm{AUCcompound}-\mathrm{AUCvehicle}}{\mathrm{AUCsham}-\mathrm{AUCvehicle}}\times 100$$

### 4.5. Thermal Hyperalgesia

Thermal hyperalgesia was analyzed by the Hargreaves method [42]. Polyalthic acid, naproxen or saline solution with 0.05% of Tween-80 was administered, and 30 min later, each rat was placed in an individual Plexiglas cubicle to become acclimated to its environment for 30 min. Subsequently, carrageenan was injected (as indicated in the previous section) into the left hind paw. The animals were immediately returned to their cubicles situated on top of the glass surface of a thermal box stimulator, which was maintained at a temperature of 30 ± 1 °C. The heat source was a light bulb, which was manipulated manually to affect the plantar surface of the injected paw. The latency time for paw withdrawal was observed, setting a cutoff point of 20 s to avoid tissue lesion. This evaluation was performed at the basal time and then every 30 min for 6 h. The percentage of MPE was calculated using the following equation:$$\%\mathrm{MPE}=\frac{\mathrm{AUCtreatment}-\mathrm{AUCvehicle}}{\mathrm{AUCcut}-\mathrm{off}}\times 100$$

### 4.6. Isobologram

Since studies have shown that an isobolographic analysis is the best method for determining the type of interaction between antinociceptive drugs [43,44], it was used presently for the combination treatment of polyalthic acid and naproxen. First, the dose resulting in a 50% antihyperalgesic effect (ED_50_) of the individual treatments of polyalthic acid and naproxen were calculated from each drug’s dose–response curve. The proportion of the combination (1:1, being 0.5 polyalthic acid and 0.5 naproxen) was the basis for calculating the theoretical point of the combined treatment. Five different doses of the combined treatment were administered: polyalthic acid ED_50_ + naproxen ED_50_, followed by one-half, one-fourth, one eighth and one-sixteenth of this dose (for details, see Table 1). The dose–response curve of the combined treatment was constructed and examined to find the experimental ED_50_. To establish the type of interaction between polyalthic acid and naproxen, the theoretical and experimental ED_50_ were compared with the Student’s *t*-test, and statistical differences were considered. Additionally, the interaction index (γ) was calculated by the following equation:$$(\gamma )=\frac{{\mathrm{ED}}_{50}\;\mathrm{of}\;\mathrm{combination}\;\left(\mathrm{experimental}\right)}{{\mathrm{ED}}_{50}\;\mathrm{of}\;\mathrm{combination}\;\left(\mathrm{theorical}\right)}\times 100$$

The interaction index indicates what portion of the ED_50_ of the individual drugs accounts for the corresponding ED_50_ of the combination. A value near 1 corresponds to an additive interaction, >1 to an antagonistic interaction, and <1 to a synergistic interaction.

### 4.7. Carrageenan-Induced Paw Edema

Edema resulted from the injection of 50 µL of 1% carrageenan (10 mg/mL, type IV, Lambda, dissolved in 0.9% saline solution) into the left hind paw, as previously described [45]. A plethysmometer (Plethysmometer 7150, Ugo Basile, Italy) was utilized to measure the volume (mL) displaced by the paw edema, a procedure performed at the basal time and then at every hour for 6 h. The percentage of the anti-inflammatory effect was calculated by the following equation:$$\%\mathrm{Anti}\text{-}\mathrm{inflammatory}\;\mathrm{effec}=\frac{\mathrm{Control}\;\mathrm{volume}-\mathrm{Test}\;\mathrm{volume}}{\mathrm{Control}\;\mathrm{volume}}\times 100$$

### 4.8. Experimental Design

Polyalthic acid (at increasing doses) or the vehicle were orally administered to independent groups of rats. Distinct series of doses were established for the different effects being analyzed: 56, 100, 178 and 320 mg/kg for antinociception; 100, 128, 320 and 562 mg/kg for the antiallodynic effect; and 10, 32, 56, 100 and 178 mg/kg for antihyperalgesic and anti-inflammatory activity. To explore the mechanism of action involved in the antinociceptive effect, the rats were intraperitoneally administered a non-selective antagonist of opioid receptors (naltrexone, 1 mg/kg), a non-selective antagonist of 5HT1,5 receptors (methiothepin, 0.1 mg/kg), or a selective inhibitor of nitric oxide-sensitive guanylyl cyclase (ODQ, 2 mg/kg). The antagonists and inhibitors were injected 10 min previous to administering polyalthic acid (320 mg/kg).

Before appraising the effects of the combined treatment, it was necessary to determine the antihyperalgesic activity of naproxen. Rats were orally administered this NSAID (at 1, 3.2, 5.6, 10 and 32 mg/kg) or the vehicle. Thus, the ED_50_ of the antihyperalgesic effect could be calculated for the individual treatments of naproxen and polyalthic acid (20.1 ± 4.8 and 101.6 ± 30.9 mg/kg, respectively). Evaluation was carried out for five distinct doses of the combined treatment (Polyalthic acid: naproxen = 50.8:10.0, 25.4:5.0, 12.7:2.5, 6.4:1.3 and 3.2:0.6; see Table 1) and the vehicle.

### 4.9. Statistical Analysis

All data are expressed as the mean ± SEM (*n* = 6). Differences between the values for the distinct groups were examined with one-way analysis of variance (ANOVA) followed by the Student–Newman-Keuls test. Differences between the theoretical and experimental ED_50_ of the combined treatment were analyzed with the Student’s *t*-test, considering significance at *p* ≤ 0.05.

## 5. Conclusions

This is the first report on the therapeutic potential of polyalthic acid for treating pain. In addition to the present evidence of its antinociceptive activity, provided with different pain models, anti-inflammatory activity was also shown. Furthermore, opioid receptors and guanylate cyclase were involved in its antinociceptive mechanism of action, which generated a synergistic effect in the combination treatment with naproxen. These results highlight the importance of further research on labdane-type diterpenes to develop drugs and combination treatments for pain and inflammation.

## Figures and Tables

**Figure 1 molecules-26-02921-f001:**
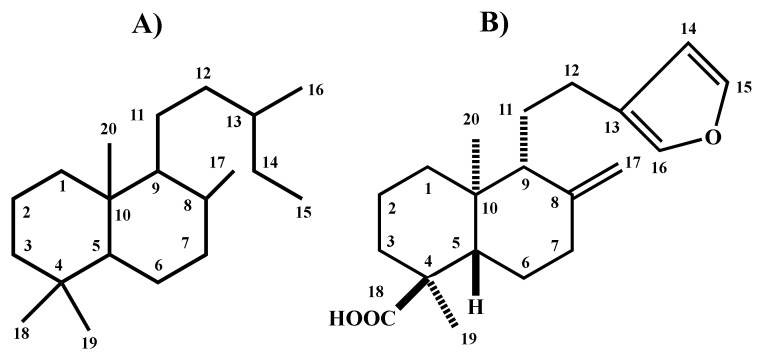
Representation of the following structures: (**A**) labdane-type diterpenes and (**B**) polyalthic acid.

**Figure 2 molecules-26-02921-f002:**
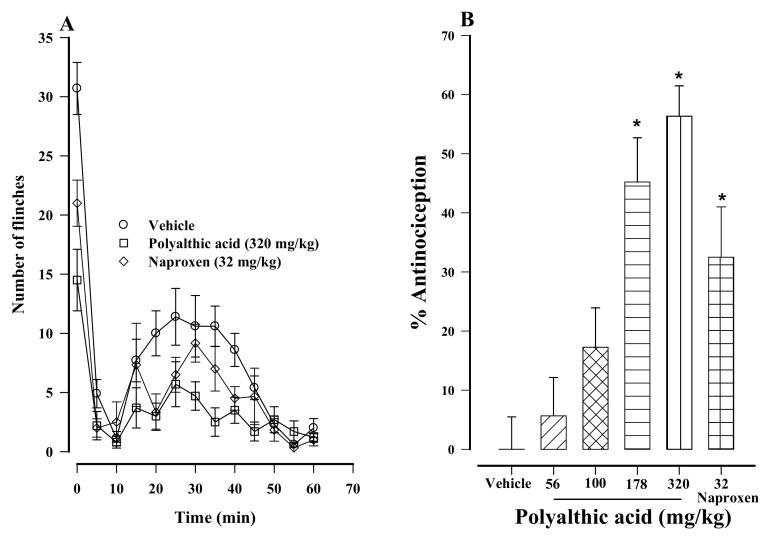
(**A**) Time course of the antinociceptive activity produced by the oral administration of polyalthic acid (320 mg/kg) and naproxen (32 mg/kg) to rats evaluated by the 1% formalin test. Data are expressed as the mean number of flinches ± SEM (*n* = 6). (**B**) Dose–response of orally administered polyalthic acid (56–320 mg/kg) during the second phase of the 1% formalin test. Data are expressed as a percentage of antinociception, considering the maximum possible effect. * *p* < 0.0001 vs. the vehicle group, according to the analysis with one-way ANOVA followed by the Student–Newman–Keuls test.

**Figure 3 molecules-26-02921-f003:**
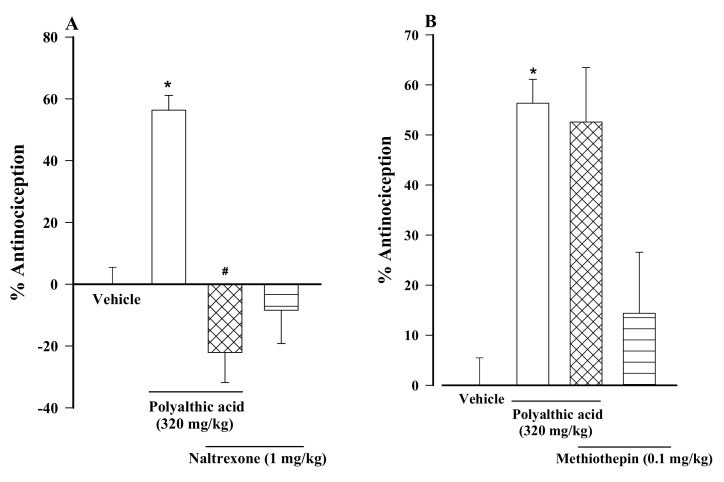
Evaluation of the participation of opioid receptors (with naltrexone), 5HT1,5 receptors (with methiothepin), and cyclic GMP (with ODQ) in the antinociceptive activity of polyalthic acid (320 mg/kg) in rats submitted to 1% formalin test. Naltrexone, methiothepin and ODQ were intraperitoneally injected into the animals 10 min before administering polyalthic acid. The effect is shown of the pretreatment with (**A**) naltrexone (1 mg/kg, i.p.), (**B**) methiothepin (0.1 mg/kg, i.p.) and (**C**) ODQ (2 mg/kg, i.p.). Data are expressed as the mean number of flinches ± SEM (*n* = 6). * *p* < 0.0001 vs. the vehicle group; ^#^
*p* < 0.001 vs. polyalthic acid in the absence of pretreatment; significance was determined by one-way ANOVA followed by the Student-Newman-Keuls test.

**Figure 4 molecules-26-02921-f004:**
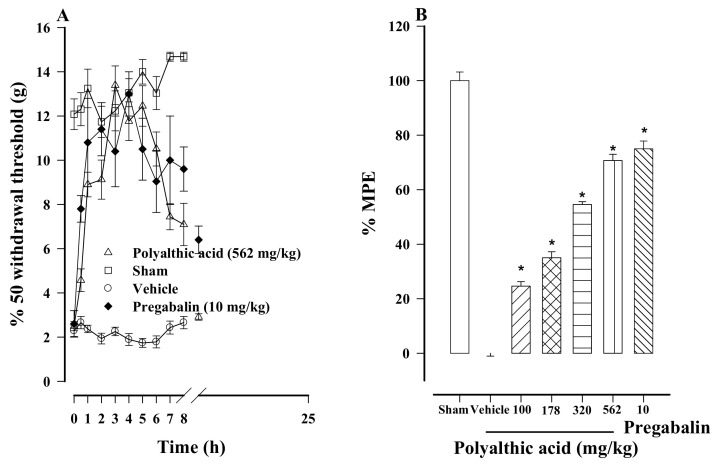
(**A**) Temporal course of the antiallodynic activity afforded by the oral administration of polyalthic acid (562 mg/kg) and pregabalin (10 mg/kg) to rats with an L5/L6 spinal nerve ligation. Data are expressed as the intensity of stimulus required to induce a response in 50% of the applications. (**B**) Dose-dependent antiallodynic effect of polyalthic acid administered at 100 to 562 mg/kg. Data are expressed as a percentage of activity against allodynia, considering the maximum possible effect (% MPE). Bars represent the mean ± SEM (*n* = 6). * *p* < 0.0001 vs. the vehicle group, as determined by one-way ANOVA and the Student–Newman–Keuls test.

**Figure 5 molecules-26-02921-f005:**
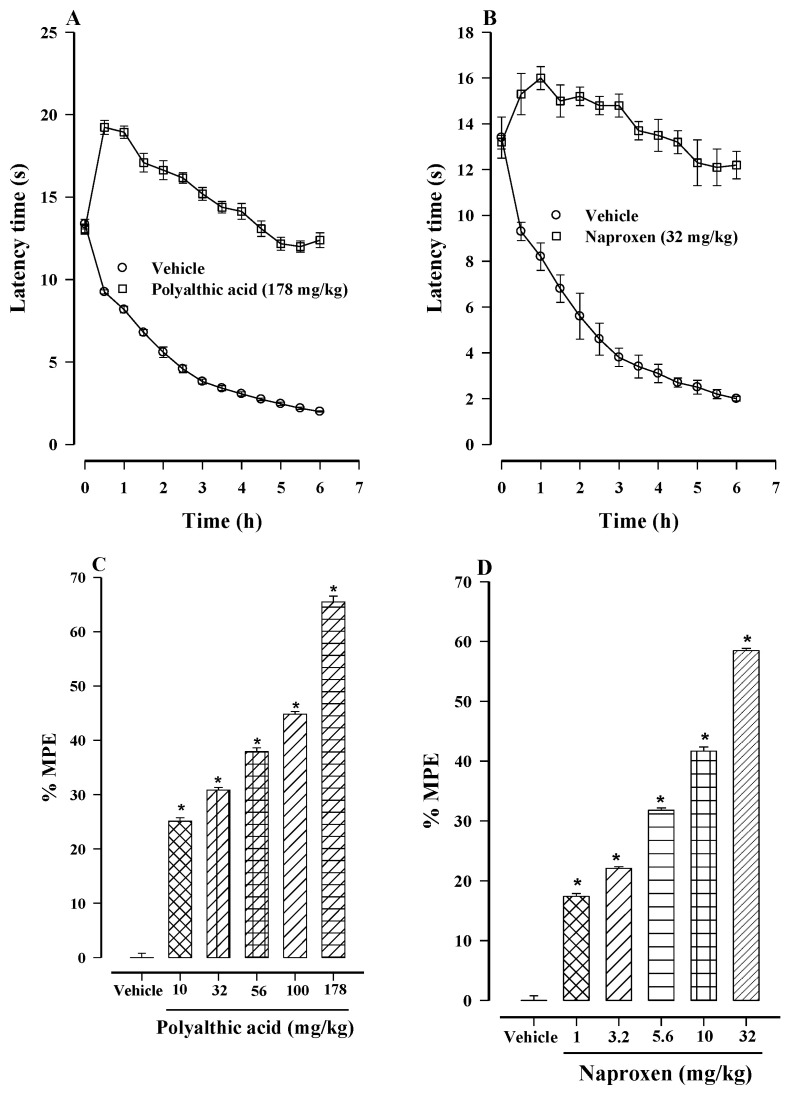
Temporal course of the antihyperalgesic activity of (**A**) polyalthic acid (178 mg/kg) and (**B**) naproxen (32 mg/kg). Hyperalgesia was induced by injecting 1% carrageenan into the left hind paw of rats. Data are expressed as latency time (in seconds) to withdraw the swollen paw from the heated surface. A dose-dependent effect was produced by orally administering: (**C**) polyalthic acid at 10 to 178 mg/kg and (**D**) naproxen at 1 to 32 mg/kg. Data are expressed as a percentage of activity against hyperalgesia, considering the maximum possible effect (% MPE). The ED_50_ was calculated by linear regression. Bars represent the mean ± SEM (*n* = 6). * *p* < 0.0001 vs. the vehicle-treated rats injected with carrageenan, as determined by one-way ANOVA and the Student–Newman–Keuls test.

**Figure 6 molecules-26-02921-f006:**
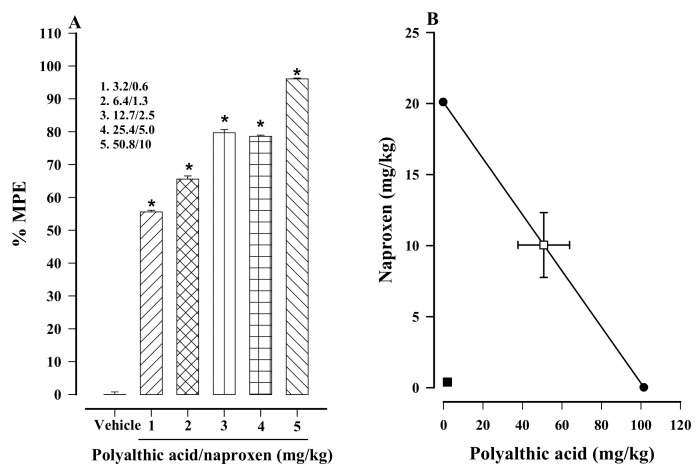
(**A**) Dose-dependent effect resulting from administering each of the five doses of the combination treatment on carrageenan-induced hyperalgesia in rats. Data are expressed as a percentage of activity against hyperalgesia, considering the maximum possible effect (% MPE). The effective dose affording 50% of the hyperalgesic activity compared to the vehicle-treated control (ED_50_) was calculated by linear regression. (**B**) Isobolographic portrayal of the interaction of polyalthic acid and naproxen in the combination treatment. Bars represent the mean ± SEM (*n* = 6). * *p* < 0.0001 vs. the vehicle-treated 1% carrageenan group, as determined by one-way ANOVA and the Student–Newman–Keuls test.

**Figure 7 molecules-26-02921-f007:**
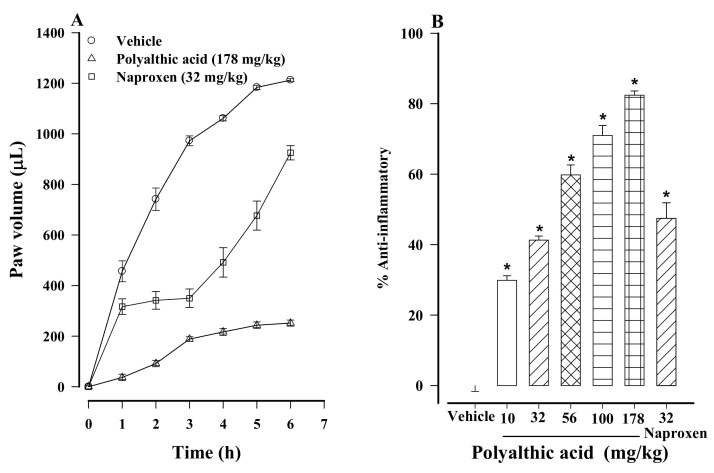
(**A**) Temporal course of the anti-inflammatory activity generated by the oral administration of polyalthic acid (178 mg/kg) to rats with 1% carrageenan-induced inflammation. Data are expressed as the volume of the edema on the left hind paw. (**B**) Dose-dependent anti-inflammatory effect elicited by polyalthic acid (10–178 mg/kg). Data are expressed as a percentage of activity against inflammation. Bars represent the mean ± SEM (*n* = 6). * *p* < 0.0001 vs. the vehicle-treated rats injected with carrageenan (1%), as determined by one-way ANOVA and the Student–Newman–Keuls test.

**Table 1 molecules-26-02921-t001:** The following doses of the combination treatment (Polyalthic acid + naproxen) were orally administered to rats evaluated with carrageenan-induced hyperalgesia. The results were used to construct a dose–response curve.

Combination	Oral Dose (mg/kg)		
	Polyalthic Acid	Naproxen	Total Dose in the Combination
1	3.2	0.6	3.8
2	6.4	1.3	7.7
3	12.7	2.5	15.2
4	25.4	5.0	30.4
5	50.8	10	60.8

**Table 2 molecules-26-02921-t002:** The experimental ED_50_ of naproxen and polyalthic acid, along with the theoretical and experimental ED_50_ of the combination treatment.

Drug	ED_50_ (mg/kg)
Naproxen	20.1 ± 4.8
Polyalthic acid	101.6 ± 30.9
Theoretical ED_50_ of the polyalthic acid–naproxen combination	60.9 ± 15.7 CI (32.4–114.3)
Experimental ED_50_ of the polyalthic acid-–naproxen combination	2.4 ± 0.02 * CI (0.9–6.6)
Interaction index (γ)	0.04 ± 0.02 CI (0.02–0.10)

ED_50_ = effective dose resulting in a 50% reduction in the hyperalgesic response found with the vehicle-treated control; CI = confidence interval. Data are expressed as the mean ± SEM (*n* = 8). * *p* < 0.05 vs. the theoretical ED_50_ of the combination treatment, as determined by the Student’s *t*-test.

## Data Availability

The data presented in this study are available on request from the corresponding author.

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
