# Peer review of "Evaluation of the Antinociceptive, Antiallodynic, Antihyperalgesic and Anti-Inflammatory Effect of Polyalthic Acid"

_molecules, 2021, doi:10.3390/molecules26102921_

Round 1

Reviewer 1 Report

Authors improved the manuscript. They added some clarifications and more references

Reviewer 2 Report

No suggestions.

This manuscript is a resubmission of an earlier submission. The following is a list of the peer review reports and author responses from that submission.

Round 1

Reviewer 1 Report

  • Line 48 – For the below statement add more references

 “Although NSAIDs that are selective cyclooxygenase-2 inhibitors (coxibs) reduce the incidence of gastrointestinal complications, they have been linked to kidney problems and an increased risk of cardiovascular complication”

  • Authors talk about medicinal plants that potentially can be applied for the pain management, but don’t show any structures of active metabolites.

Add chemical (representative) structures of labdane-type diterpenes and polyalthic acid

  • Line 57- Authors claim that anti- inflammatory function of polyalthic acid was not studied, however I did find the following reference:

Evid Based Complement Alternat Med. 2018; 2018: 5617234.

Published online 2018 Jan 23. doi: 10.1155/2018/5617234

  • Copaiba oil contains polyalthic acid, and there are some studies that need to be addressed in the discussion.

For example: Hebert et al Integr Med (Encinitas). 2017 Apr; 16(2): 40–42.

  • What is the rationale to perform experiment only with female animals? If only female were available for the study, authors need to add it a study limitation and also make a discussion what are projection/expectation of the same experiment in male population of animals

  • There is a poor design of the anti- inflammatory action of polyalthic acid. I would expect to see some circulating markers levels and their correlation to the polyalthic acid dosage. If such a data was not collected during the experiment, that deficiency should be addressed in the body of the manuscript
  • Add legends to all graphs

Reviewer 2 Report

  • Please give more date about how you obtain, analyse and control the purity of polyalthic acid used, more data about the plant.
  • Is it use in popular medicine like analgesic or anti-inflammatory remedy? (bibliography)
  • There are some data about polyalthic acid toxicity? How do you establish the doses?
  • How you prepared polyalthic acid for oral administration?
  • For Fig 1A, you must have a negative control (vehicle) and a lot treated with polyalthic acid (both treated with formalin). The same for figure 4A, 5A, 5B.
  • It is not clear why you use different range of doses for each experiment
  • Normally, in an experimental study you must use a positive control (an anti-inflammatory drug, an analgesic drug to compare). Please complete with a positive control lot
  • The first column in 4A represents the vehicle, not the carrageenan. Same for 5C and 5D
  • Table 1, last column is not correct. We cannot add the two values like that.
  • The references are old. Can you supplement with references from recent years?
  • Why you choose naproxen to study the synergism?
  • 209-please write L-arginine
  • Please explain why you presume that can be a pharmacokinetics interaction between naproxen and polylactic acid? This explanation in my opinion is not the best “According to one report, 14 days of continuous exposure of patients to naproxen in a clinical trial did not induce the metabolism of enzymes [30]. The same lack of metabolism of enzymes was observed after exposure to an overdose of naproxen [31]. On the other hand, no information was found in the literature about the pharmacokinetics of polyalthic acid. However, a pharmacokinetic interaction between polyalthic acid and naproxen is not discarded, and 244 future research is needed in this regard. “
  • Please give more explanation on discussions about antiallodynic, anti inflammatory effect of polyalthic acid.
  • Please add some conclusions (the utility for therapy)

Reviewer 3 Report

The submitted manuscript presents relevant data on the antinociceptive, antiallodynic, anti-inflammatory and 3 antihyperalgesic and polyalthic acid. Authors should reread the manuscript in order to minimize punctuation errors. Identify the possibility of reducing the number of tables presented in the work. Indicate at work the origin or obtaining of polyalthic acid, since it is a natural product. Is table 1 properly represented according to Molecules instructions?